# Effect of Calcination Temperature on the Structure, Crystallinity, and Photocatalytic Activity of Core-Shell SiO$_2$@TiO$_2$ and Mesoporous Hollow TiO$_2$ Composites

Ning Fu [1,2], Hongjin Chen [1], Renhua Chen [1], Suying Ding [1] and Xuechang Ren [1,2,*]

[1] School of Environmental and Municipal Engineering, Lanzhou Jiaotong University, Lanzhou 730070, China
[2] Key Laboratory of Yellow River Water Environment in Gansu Province, Lanzhou Jiaotong University, Lanzhou 730070, China
* Correspondence: rxchang1698@hotmail.com

**Abstract:** TiO$_2$ and core–shell SiO$_2$@TiO$_2$ nanoparticles were synthesized by sol-gel process at different calcination temperatures. Mesoporous hollow TiO$_2$ composites were prepared by etching SiO$_2$ from SiO$_2$@TiO$_2$ nanoparticles with alkali solution. X-ray diffraction (XRD), Scanning electron microscope (SEM),Transmission electron microscope (TEM), and N$_2$ adsorption–desorption isotherms, and Roman and Diffuse reflectance spectroscopy (DRS) were employed to characterize the synthesized materials. The effects of different calcination temperatures on the morphology, crystallinity, phase composition, and photocatalytic activity of the prepared materials were investigated in detail. It was found that the calcination temperature altered the phase structure, crystallinity, morphology, specific surface area, and porous structure. Additionally, it was verified that SiO$_2$ could inhibit the transfer of TiO$_2$ from anatase phase to rutile phase under high temperature calcination (850 °C). The hollow TiO$_2$ calcined at 850 °C showed the highest photocatalytic efficiency of 97.5% for phenol degradation under UV irradiation.

**Keywords:** TiO$_2$; core–shell SiO$_2$@TiO$_2$; hollow TiO$_2$; calcination temperature





## 1. Introduction

Titanium dioxide (TiO$_2$), as one of the most promising photocatalyst materials, has many advantageous properties, such as low toxicity, high photocatalytic activity, low cost, and high chemical stability [1–3]. TiO$_2$ can generate electron-hole pairs and react with H$_2$O/O$_2$ to generate •O$_2$/•OH radicals for degradation of organic pollutants in water [4]. However, pure TiO$_2$ has large band gap(Eg 3.0–3.2 eV), easy agglomeration, phase transformation, and high electron-hole pair recombination rate, which reduces its photocatalytic activity [5]. It has been reported that nanoscale TiO$_2$ exhibits enhanced photocatalytic activity under the morphological control of nanorods, nanotubes, core–shell nanospheres, and hollow spheres [6]. In general, these nanostructured TiO$_2$ have much higher specific surface areas than bulk TiO$_2$ with improved light utilization, which can shorten the charge carrier diffusion path length of the electron-hole pair and promote the increase in electrochemical reaction at the aqueous interface [7].

Among these controlled morphologies, core–shell nanospheres and hollow spheres have attracted wide attention due to their improved physical and chemical properties, such as higher surface area, low density, controlled morphology, surface permeability, and improved light-trapping effect [8]. It has been reported that SiO$_2$ can be used as solid core and TiO$_2$ as shell to synthesis core–shell SiO$_2$@TiO$_2$ nanospheres for photocatalytic degradation of dyes and other organic pollutants [9]. Similarly, hollow TiO$_2$ spheres can be synthesized by "hard templating" method, in which a SiO$_2$ core is etched with an alkaline solution [10].

During the process of synthesizing the core–shell $SiO_2@TiO_2$ nanospheres and hollow $TiO_2$ spheres, calcination process is usually used to form photocatalyst to promote phase transformation, thermal decomposition and crystallinity [11]. Therefore, applying calcination to the formation of doped $TiO_2$ can also improve its photocatalytic activity, morphology, surface area, crystallinity, and the photoabsorption of the photocatalyst.

For $TiO_2$ photocatalyst, phase composition, surface area, and crystallinity are three crucial factors that affect the photocatalytic activity [4]. It is well known that $TiO_2$ has three crystal forms of phase: brookite, anatase, and rutile. Among them, brookite is unstable. Anatase has higher reduction potential and lower recombination rate of electron-hole pairs, and can also produce more oxygen vacancies to capture electrons; thus, being identified as the most active phase from a photocatalytic point. Rutile has a lower band-gap energy (3.0 eV) than anatase with more stable crystalline structure, resulting in easy recombination of electrons-holes and almost no photocatalytic activity [8]. At the optimum calcination temperature [12], a well crystallized $TiO_2$ anatase phase material has large surface area and small grain size, which is the preferred form of $TiO_2$ based photocatalyst. However, the perfectly crystallized hollow $TiO_2$ often conflicts with the realization of large surface area, because the increased calcination temperature used for $TiO_2$ crystallization may lead to further $TiO_2$ growth or agglomeration, ultimately reducing the specific surface area [13]. Generally, in the process of preparing hollow $TiO_2$ by alkaline etching method, the phase structure of hollow $TiO_2$ is unstable and easy to be destroyed. Additionally, the hollow $TiO_2$ cannot form a uniform and complete hollow structure if the calcination temperature is too low; while if the calcination temperature is too high, the core–shell $SiO_2@TiO_2$ is easy to be destroyed and the hollow $TiO_2$ will form rutile phase, which is not conducive to improve the photocatalytic activity [14]. Therefore, how to control the appropriate calcination temperature to form hollow $TiO_2$ with complete morphology and stable phase composition is very important for the preparation of hollow $TiO_2$. At present, few articles focus on systematic research and comparison of the influence of different calcination temperatures on the phase change structure and morphology characteristic of $SiO_2@TiO_2$ nanospheres and mesoporous hollow $TiO_2$ nanoparticles [4,7,14].

In this paper, the $TiO_2$, core–shell $SiO_2@TiO_2$ and mesoporous hollow $TiO_2$ were prepared under different calcination temperatures. The effects of calcination temperature on microstructure, crystallinity, phase composition, and photocatalytic activity of $TiO_2$, core–shell $SiO_2@TiO_2$ and mesoporous hollow $TiO_2$ were investigated. Moreover, the correlation between calcination temperatures and microstructure, crystallinity and photocatalytic performance was discussed comparatively.

## 2. Experimental Procedure

### 2.1. Experiment Materials

Tetraethyl orthosilicate (TEOS), tetrabutyl titanate (TBT), ethanol, phenol, methylene blue (MB), and ammonia were purchased from Sinopharm chemical reagent company (China, Shanghai). None of the agents were purified.

### 2.2. Synthesis of $SiO_2$ Core Spheres

The $SiO_2$ core spheres were synthesized using the Stöber method according to the reported paper [15]. First, 10 mL TEOS was added to 90 mL ethanol. After continuous stirring, solution A was formed. At the same time, 20 mL water and 10 mL aqueous ammonia were added to 70 mL ethanol. After vigorous stirring, solution B was formed. Then, solution A and solution B were mixed under continuous stirring for 3 h at 40 °C. Finally, the products were centrifuged at 4000 rpm and washed twice with methanol and once with water. The final obtained products were dried at 70 °C for at least 20 h.

### 2.3. Synthesis of Core-Shell $SiO_2@TiO_2$ Nanospheres

The core–shell $SiO_2@TiO_2$ nanospheres were prepared by sol-gel process according to the literature [16]. First, 0.3 g $SiO_2$ was sonicated in 100 mL ethanol for 30 min to obtain

solution A. 4 mL TBT was added into 100 mL ethanol to obtain solution B. Then, solution B and 1.5 mL aqueous ammonia were added to solution A under vigorous stirring at 60 °C for 3 h. The resulting precipitates were centrifuged at 8000 rpm, washed twice with ethanol and once with water. Finally, these obtained products were dried at 70 °C for at least 20 h. Additionally, through the same procedures, unsupported $TiO_2$ was also synthesized using 4 mL TBT by the same above procedures without $SiO_2$ in the mixture.

### 2.4. Synthesis of $TiO_2$ and $SiO_2@TiO_2$ Nanoparticles at Different Calcination Temperatures

The prepared $TiO_2$ was calcined at 450 °C, 650 °C, 850 °C, and 1050 °C for 2 h, and the heating rate was 10 °C/min. The synthesized $SiO_2@TiO_2$ core–shell nanospheres were calcined for 2 h at 450 °C, 550 °C, 650 °C, 750 °C, 850 °C, 950 °C, and 1050 °C with the calcination heating rate of 10 °C/min, respectively. All the samples were synthesized in air atmosphere during calcination.

### 2.5. Synthesis of Mesoporous Hollow $TiO_2$ Nanoparticles

An amount of 0.5 g core–shell $SiO_2@TiO_2$ nanoparticles calcined at different temperatures were dispersed in 60 mL water under ultrasonication for 30 min. Subsequently, 3 mL 2.5 M NaOH solution was added into the solution to etch the $SiO_2$ core form $SiO_2@TiO_2$ nanoparticles under vigorous stirring at 30 °C for 6 h. Then, the etched hollow $TiO_2$ nanoparticles were isolated by centrifugation at 8000 rpm, washed twice with ethanol and once with water. Finally, the obtained hollow $TiO_2$ nanoparticles were dried at 70 °C for at least 20 h.

### 2.6. Characterization

The morphology of samples was characterized by scanning electron microscope (SEM, JSM-6701F, Japan), the sample was dispersed in ethanol solution and dropped onto a copper sample table. After spraying gold three times to increase conductivity, it was tested using SEM. Transmission electron microscopy (TEM, TECNAI G2,USA) was also used to test the morphology of samples with copper mesh as support film. The phases of the samples were performed on an X-ray diffraction (XRD), which was carried out by a RINT 2000 equipment with Cu K$\alpha$ radiation (40 Kv/30 mA). The pore structure and Brunauer–Emmett–Teller (BET) specific surface area of the composite particles were determined from nitrogen gas absorption-desorption using an ASAP 2020 instrument. Raman spectroscopy was characterized by a Raman spectrometer (Renishaw 2000L, Britain, UK). UV-Vis diffuse reflectance spectra (UV-vis DRS) was tested by a UV-Vis spectrophotometer (Lambda 950) with a wavelength range of 200–800 nm and a reflectance standard of $BaSO_4$. The photodegradation of phenol was detected by UV spectrophotometer(UV-3100).

### 2.7. Measurement of Photocatalytic Performance

The photocatalytic performance of the samples was evaluated by measuring the photodegradation of phenol in a reactor under UV light irradiation (500 W Hg lamp) for 180 min. In general, 75 mg of prepared photocatalysts was dispersed in 300 mL phenol aqueous solution with an initial concentration of 20 mg/L and stirred in the dark for 30 min to reach the adsorption–desorption equilibrium, then carried out under UV light. An amount of 4 mL solution was collected from the reactor at different irradiation intervals and the suspended particles were separated by centrifugation. The concentration of phenol was determined at 510 nm by colorimetric method of 4-amino antipyrine.

## 3. Results and Discussion

### 3.1. Structure and Characterization

#### 3.1.1. SEM and TEM

The morphology and crystallinity of $TiO_2$, $SiO_2@TiO_2$, and hollow $TiO_2$ were characterized and compared by TEM and SEM, as shown in Figures 1–4. Figure 1 showed the morphology and crystallinity of $SiO_2@TiO_2$ prepared at different calcination tempera-

tures. The overall morphology of SiO$_2$@TiO$_2$ nanoparticles was spherical and the TiO$_2$ was loaded on the surface of SiO$_2$, thus forming a core–shell structure with a thickness of about 40–70 nm. When the calcination temperatures increased from 450 °C to 1050 °C, the shell surface TiO$_2$ particle size of SiO$_2$@TiO$_2$ gradually increased and the core–shell structure became more obvious as shown in Figure 1a–d. In Figure 2a–d, the crystal structures of SiO$_2$@TiO$_2$ were further characterized by HRTEM. The images showed that the particle size of TiO$_2$ gradually increased and the thickness of TiO$_2$ shell structure decreased with the increase in calcination temperatures. The TiO$_2$ shell sizes of SiO$_2$@TiO$_2$ were about 7 nm, 10 nm, 15 nm, and 30–50 nm when the calcination temperatures were 450 °C, 650 °C, 850 °C, and 1050 °C, respectively. Compared with the size of individual TiO$_2$ particles prepared at different calcination temperatures, the particle size of TiO$_2$ from SiO$_2$@TiO$_2$ was smaller than that of pure TiO$_2$, which was consistent with the XRD characterization results of SiO$_2$ inhibiting the nucleation and growth of TiO$_2$ [17].

Figure 3 showed SEM images of hollow TiO$_2$ prepared at different calcination temperatures. As shown in Figure 3, hollow TiO$_2$ prepared at 450 °C and 650 °C had more obvious hollow structure than the hollow TiO$_2$ prepared at 850 °C and 1050 °C. At the same time, with the increase in calcination temperature, TiO$_2$ on the surface of SiO$_2$ is easy to fall off and form agglomerated particles in the process of calcination and alkali corrosion. This was because with the increase in calcination temperature, the structure of SiO$_2$@TiO$_2$ core–shell photocatalyst became more stable, the binding force of Si-O-Ti bond was enhanced, and it was more difficult to remove SiO$_2$ with alkaline solution. With the increase in the calcination temperature, the TiO$_2$ particle size on the surface of SiO$_2$ became larger during the calcination process, which was easy to fall off and to form agglomerated particles. At the same time, the increase in calcination temperatures would enhance the binding force of Si-O-Ti bond and make the SiO$_2$@TiO$_2$ core–shell structure more stable; thus, it was more difficult to etch SiO$_2$ with alkaline and the hollow structure morphology was not obvious [18].

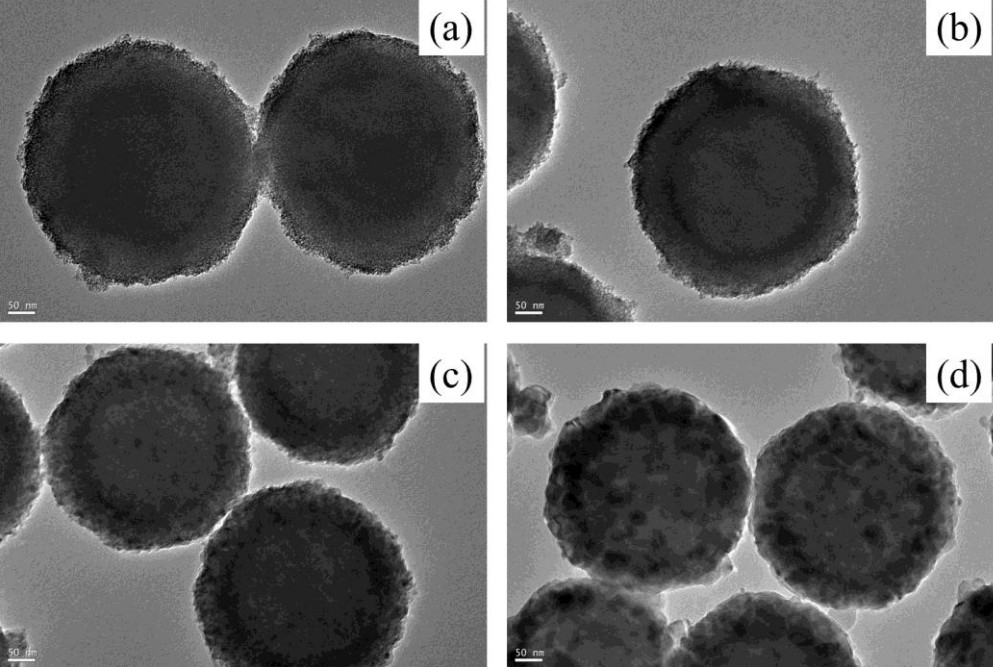

**Figure 1.** TEM images of SiO$_2$@TiO$_2$ prepared at different calcination temperatures (**a**) 450 °C, (**b**) 650 °C, (**c**) 850 °C, (**d**) 1050 °C.

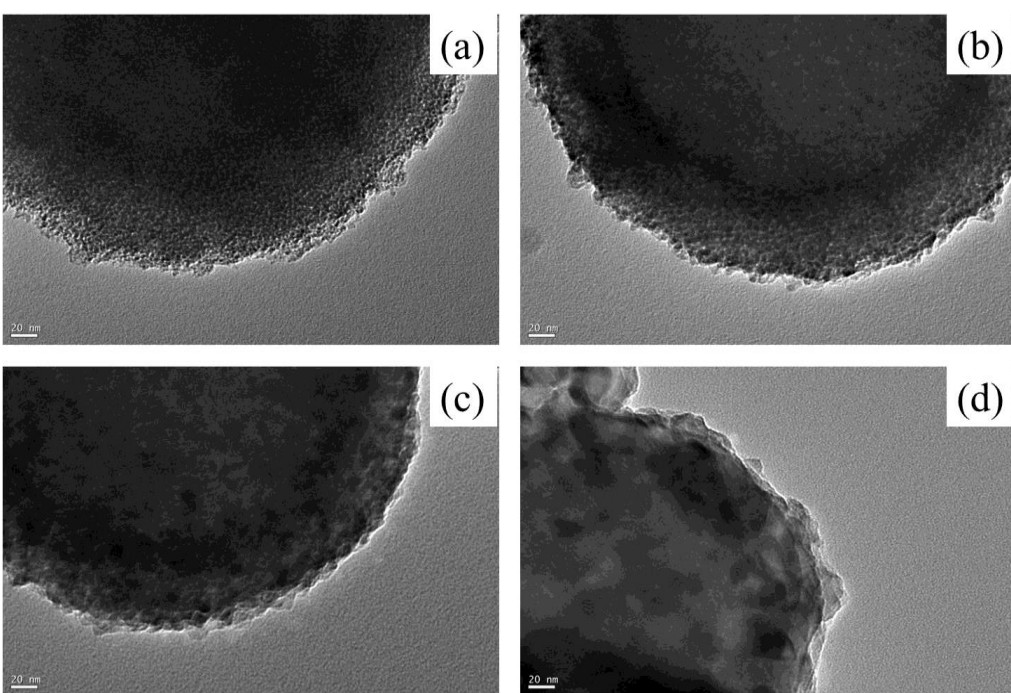

**Figure 2.** HRTEM images of SiO$_2$@TiO$_2$ prepared at different calcination temperatures (**a**) 450 °C, (**b**) 650 °C, (**c**) 850 °C, (**d**) 1050 °C.

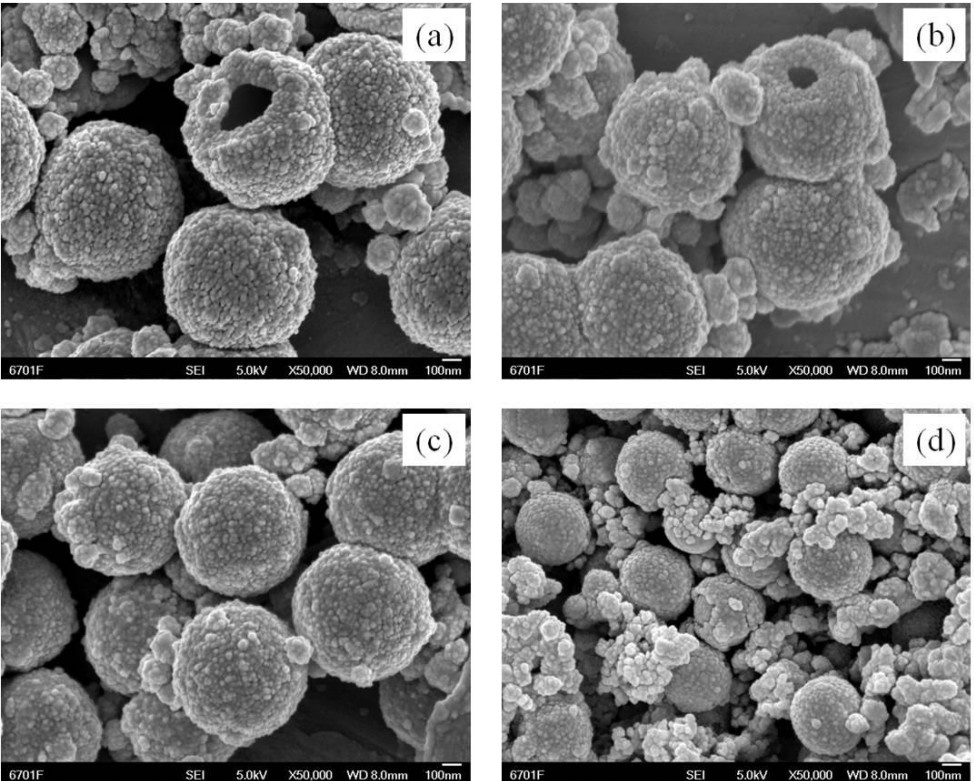

**Figure 3.** SEM images of hollow TiO$_2$ prepared at different calcination temperatures (**a**) 450 °C, (**b**) 650 °C, (**c**) 850 °C, (**d**) 1050 °C.

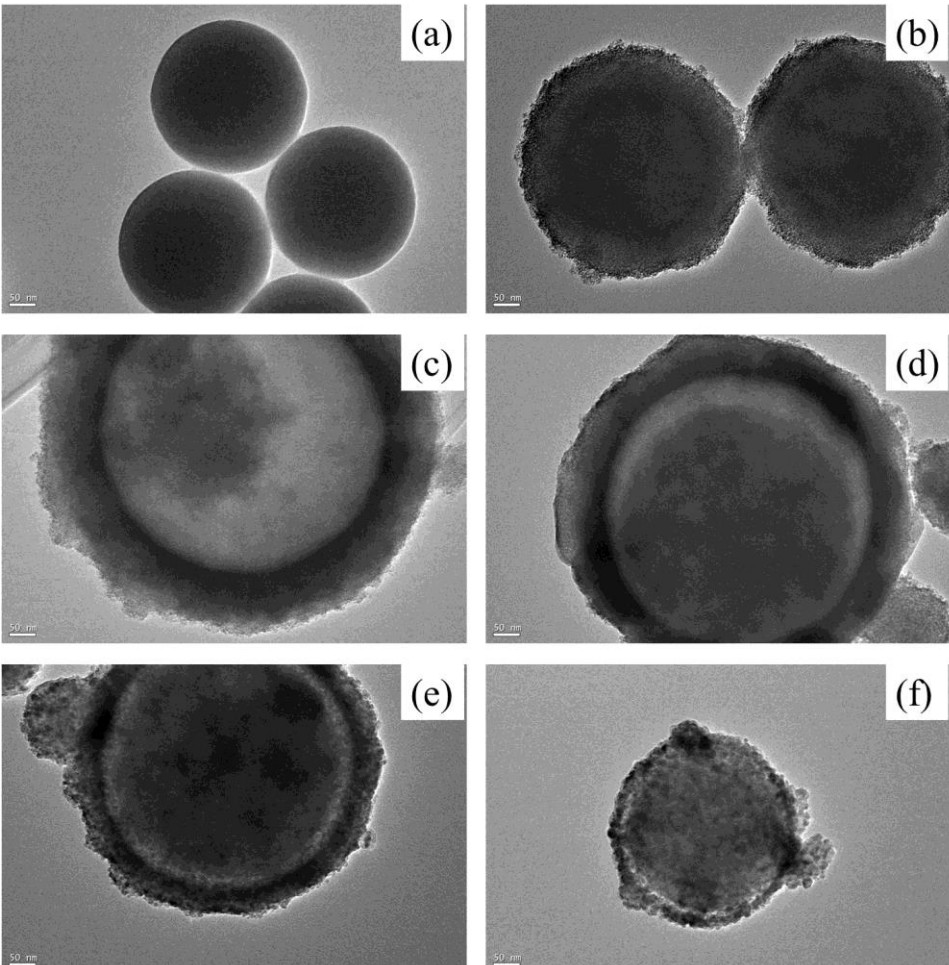

**Figure 4.** TEM images of hollow $TiO_2$ prepared at different calcination temperatures (**a**) $SiO_2$, (**b**) $SiO_2@TiO_2$ (450 °C), (**c**) 450 °C, (**d**) 650 °C, (**e**) 850 °C, (**f**) 1050 °C.

Figure 4 showed the TEM of hollow $TiO_2$ prepared at different calcination temperatures. Compared with the $SiO_2$ and $SiO_2@TiO_2$, the morphology of hollow $TiO_2$ prepared at different calcination temperatures had hollow structure. However, there were two differences in the morphology characteristics of hollow $TiO_2$ prepared at different calcination temperatures: First, with the increase in calcination temperature, the particle size of $SiO_2@TiO_2$ increased, and the crystal structure became more stable, and the $TiO_2$ network structure containing silicate became firmer [19]. Therefore, $SiO_2$ was more difficult to remove, and the hollow structure was less obvious, which was consistent with the previous SEM results. TEM results also showed that the hollow structure of $TiO_2$ prepared at 450 °C was very obvious, while the hollow structures of $TiO_2$ prepared at 650 °C, 850 °C and 1050 °C were not very obvious, and only part of $SiO_2$ was removed. Second, as the calcination temperature increased, the thickness of $TiO_2$ shell structure of $SiO_2@TiO_2$ decreased. The thickness of the hollow $TiO_2$ shell at 450 °C, 650 °C, 850 °C, and 1050 °C calcination temperature was about 70 nm, 50 nm, 40 nm, and 20 nm, respectively (Figure 4c–f), which showed that the calcination temperature directly affected the morphology and structural stability of hollow $TiO_2$ [20].

### 3.1.2. XRD

Figure 5 showed the XRD patterns of prepared $TiO_2$, $SiO_2@TiO_2$ and hollow $TiO_2$ nanoparticles, respectively. According to the XRD pattern in Figure 5a, different calcination temperatures had an obvious effect on the crystal structure of $TiO_2$. After calcination at 450 °C, $TiO_2$ crystal structure was anatase type (JCPDS No. 21-1272) with characteristic

diffraction peak $2\theta$ = 25.3°, 38.0°, and 48.2°. However, the characteristic diffraction peak intensity of anatase was weak and the peak shape was not sharp, indicating that $TiO_2$ had low crystallinity and small particle size. Compared with $TiO_2$ calcined at 450 °C, the crystal structure of $TiO_2$ calcined at 650 °C was more obvious, and the characteristic diffraction peaks of anatase were $2\theta$ = 25.3°, 38.0°, 48.2°, 54.0°, and 62.8°, corresponding to (101), (004), (200), (105), and (204) planes of anatase $TiO_2$, respectively [21]. At the same time, the characteristic diffraction peak intensity was significantly enhanced, and the peak shape was sharp, indicating that the crystal structure of $TiO_2$ anatase was more obviously stable and the crystallinity was higher after being calcined at 650 °C. When the calcination temperature increased to 850 °C, the crystal structure of $TiO_2$ shifted from anatase phase to rutile phase, and obvious characteristic diffraction peaks at 27.5°, 36.1°, 54.3°, 56.8°, and 69.1° were shown, corresponding to (110), (101), (211), (220), and (301) planes of rutile $TiO_2$ (JCPDS No. 21-1276), respectively [9]. When the calcination temperature increased to 1050 °C, the crystal structure of $TiO_2$ rutile became more obvious, and the intensity of characteristic diffraction peak also increased, indicating that the particle size of $TiO_2$ rutile phase became larger. According to the characteristic diffraction peak properties of XRD after calcination at different temperatures, the particle sizes of $TiO_2$ at different calcined temperatures (450 °C, 650 °C, 850 °C, and 1050 °C) were 14.4 nm, 28.0 nm, 57.1 nm, and 95.5 nm, respectively, calculated by Scherrer equation ($R_m = k\lambda / \beta_{1/2} \cos\theta$). The results showed that with the increase in calcined temperature, the particle size of $TiO_2$ increases gradually [7].

Figure 5b showed the XRD results of $SiO_2@TiO_2$ nanoparticles prepared at different calcination temperatures. The XRD results of $SiO_2@TiO_2$ nanoparticles calcinated at 450 °C showed amorphous structure without obvious characteristic diffraction peak. The XRD results presented the anatase crystal diffraction peak with low intensity when the calcination temperature gradually increased from 550 °C to 750 °C. As the calcination temperature rose to 850 °C, the obvious anatase diffraction peak of $SiO_2@TiO_2$ nanoparticles appeared, while pure $TiO_2$ belonged to obvious anatase with the calcination temperatures of 450 °C and 650 °C, which indicated that the nucleation and growth of $TiO_2$ crystal were significantly inhibited due to the doping of $SiO_2$, and the structural transfer of anatase phase to rutile phase was also inhibited during the synthesis process. The reported research also indicated that the doping of $SiO_2$ would inhibit the growth and phase transfer of $TiO_2$ at higher calcination temperature [22]. When the calcination temperature rose to 950 °C and 1050 °C, the phase structure of $SiO_2@TiO_2$ nanoparticles was composed of anatase and rutile homogenous structure. The crystal structure calcinated at 950 °C transferred with some rutile phase characteristic diffraction peak at 27.5° and 36.2°, but the main diffraction characteristic peak was $2\theta$ = 25.3°, which belonged to anatase. The $SiO_2@TiO_2$ nanoparticles calcinated at 1050 °C had obvious rutile phase characteristic diffraction peak at 27.5°, 36.2°, 41.4°, 54.4°, and 56.8°, and the crystal structure of anatase was significantly weakened.

Compared with the pure $TiO_2$ crystal structure of rutile at the calcination temperature of 1050 °C, $SiO_2@TiO_2$ nanoparticles still had part of anatase crystal structure at 1050 °C, which further showed that $SiO_2$ could inhibit the transfer of $TiO_2$ crystal structure from anatase to rutile phase. The main reason $SiO_2$ could inhibit the phase transfer of $TiO_2$ was that Ti-O-Si bond was formed during the synthesis of $SiO_2@TiO_2$ nanoparticles. As the calcination temperature increases, the Ti-O-Si bond became stronger, and the transfer of $TiO_2$ from anatase to rutile became more difficult, thus inhibiting the growth of $TiO_2$ particle size and crystal phase transfer [23,24].

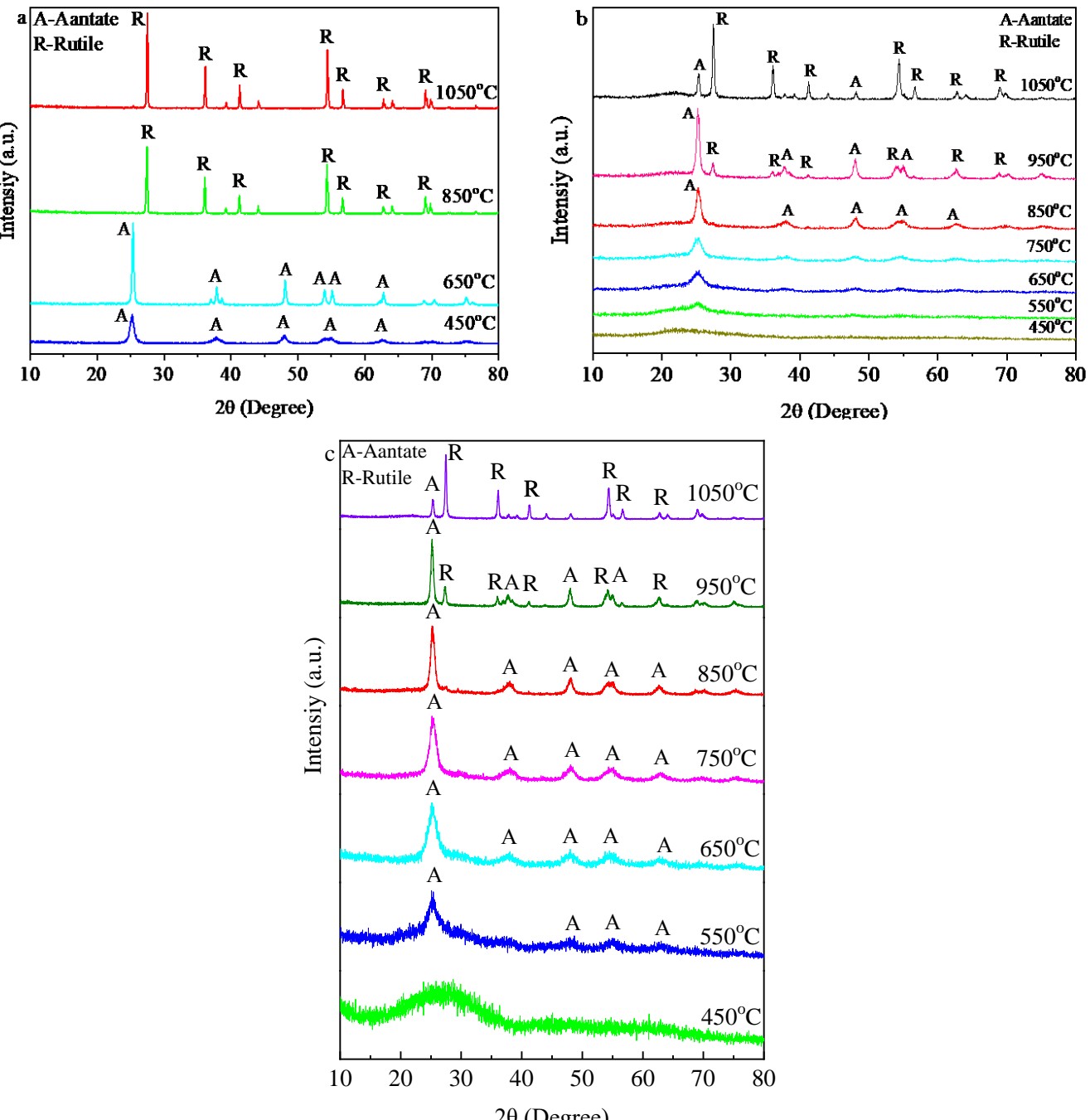

**Figure 5.** XRD spectra of (**a**) TiO$_2$, (**b**) SiO$_2$@TiO$_2$, and (**c**) hollow TiO$_2$ prepared at different calcination temperatures.

Figure 5c showed the XRD of hollow TiO$_2$ nanoparticles prepared by SiO$_2$@TiO$_2$ nanoparticles after etching with alkali. The crystal structure was basically consistent with that of SiO$_2$@TiO$_2$ nanoparticles at different calcination temperatures. The hollow TiO$_2$ nanoparticles prepared at 450 °C belonged to amorphous structure. The XRD diffraction peaks of hollow TiO$_2$ nanoparticles prepared after calcination temperature from 550 °C to 850 °C belonged to anatase crystal type. With the increase in calcination temperature, the intensity of its anatase characteristic diffraction peak increased, the peak shape became sharp, the crystallinity and the nanoparticle size also increased. The hollow TiO$_2$ nanoparticles prepared after calcination at 950 °C and 1050 °C belonged to the mixed crystal structure of anatase and rutile, which indicated that the process of alkali corrosion did not destroy the

crystalline structure of $TiO_2$. At the same time, the characteristic diffraction peak of hollow $TiO_2$ crystalline structure was more obvious and the crystallinity was higher compared with $SiO_2@TiO_2$ nanoparticles [25].

Figure 6 showed the XRD comparison diagram of $TiO_2$, $SiO_2@TiO_2$, and hollow $TiO_2$ prepared at different calcination temperatures. As can be seen from Figure 6a, when the calcination temperature was 450 °C, the crystalline structure of $TiO_2$ was anatase, while $SiO_2@TiO_2$ and hollow $TiO_2$ belonged to amorphous structure. The crystal structure of $TiO_2$, $SiO_2@TiO_2$, and hollow $TiO_2$ presented anatase crystal structure, while the characteristic diffraction peak intensity of $TiO_2$ anatase was sharper than $SiO_2@TiO_2$ and hollow $TiO_2$ when the calcination temperature was 650 °C (Figure 6b). When the calcination temperature increased to 850 °C (Figure 6c), the crystalline structure of $TiO_2$ belonged to rutile phase, but $SiO_2@TiO_2$ and hollow $TiO_2$ crystal structure was still anatase. When the calcination temperature increased to 1050 °C (Figure 6d), the crystalline structure of $TiO_2$ was rutile phase, but the crystal structure of $SiO_2@TiO_2$ and hollow $TiO_2$ belonged to the mixed crystal structure of rutile and anatase, which further indicated that $SiO_2$ inhibited the transfer of $TiO_2$ crystal structure from anatase phase to rutile phase. In addition, the XRD comparison diagram showed that the characteristic diffraction peak intensity of hollow $TiO_2$ crystal structure was higher than that of $SiO_2@TiO_2$ for high purity and crystallinity of $TiO_2$ [26].

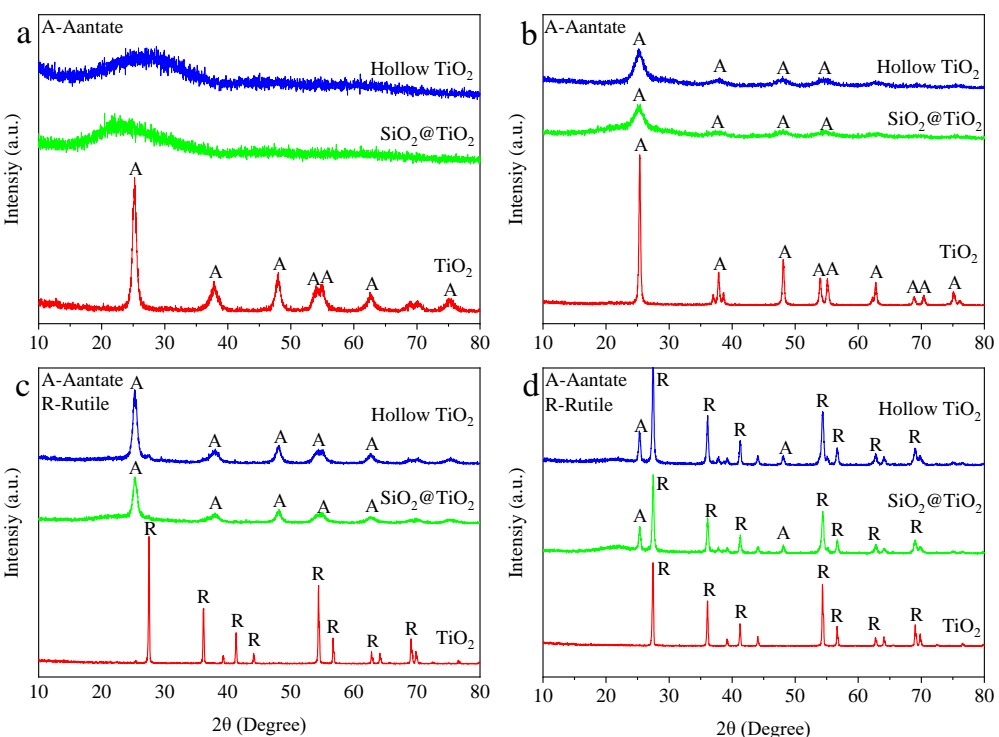

**Figure 6.** XRD spectra of prepared nanoparticles calcined at different temperatures (**a**) 450 °C, (**b**) 650 °C, (**c**) 850 °C, (**d**) 1050 °C.

### 3.1.3. BET

Figures 7 and 8 showed the $N_2$ adsorption–desorption isotherms of $TiO_2$, $SiO_2@TiO_2$, and hollow $TiO_2$. The Brunauer–Emmett–Teller (BET) method was used to calculate the surface area and pore size parameters of the synthesized samples, which were shown in Table 1. Figure 7a indicated that the $N_2$ adsorption–desorption curves calcined at 450 °C and 650 °C belonged to type IV adsorption curves with H2 type hysteresis loops and were mesoporous materials. Additionally, the $N_2$ adsorption–desorption curves of $TiO_2$ calcined at 850 °C and 1050 °C belonged to type II isotherm, which belonged to non-porous materials. As shown in Figure 7b, $N_2$ adsorption–desorption curves of $SiO_2@TiO_2$ calcined at different temperatures were quite different. The $N_2$ adsorption–desorption curves of $SiO_2@TiO_2$

calcined at 450 °C, 650 °C, and 850 °C could be categorized as type IV with H2 type hysteresis loops, belonging to mesoporous materials. The $N_2$ adsorption–desorption curve of $SiO_2@TiO_2$ calcined at 1050 °C belonged to type II isotherm and was non-porous material, which was consistent with the results of $TiO_2$ calcined at 1050 °C. As shown in Figure 7c, $N_2$ adsorption–desorption curves of hollow $TiO_2$ calcined at different temperatures belonged to type IV adsorption curves with H2 type hysteresis loops (calcined at 450 °C, 650 °C, and 850 °C) and H3 type hysteresis loop (calcined at 1050 °C). Additionally, the $N_2$ adsorption–desorption curves of hollow $TiO_2$ calcined at 650 °C and 850 °C showed more obvious H2 type hysteresis loops and more obvious mesoporous material properties.

Figure 8 showed the comparison diagram of $N_2$ adsorption–desorption curves of $TiO_2$, $SiO_2@TiO_2$, and hollow $TiO_2$ calcined at different temperatures. The results indicated that $SiO_2@TiO_2$ had more obvious mesoporous material properties with H2 type hysteresis loop when calcined at 450 °C. However, as the calcination temperatures increased to 650 °C, 850 °C, and 1050 °C, the hollow $TiO_2$ showed more obvious hysteresis ring structure, which indicated that hollow $TiO_2$ had more obvious mesoporous material properties. Compared with $TiO_2$ and $SiO_2@TiO_2$, hollow $TiO_2$ had higher specific surface area, smaller density and better dispersion [27].

Table 1 showed the specific surface area, pore size, and pore volume of the prepared photocatalysts. The results showed that the specific surface area of $TiO_2$, $SiO_2@TiO_2$, and hollow $TiO_2$ decreased with the increase in calcination temperature, which was related to the increase in particle size and the easy destruction of core–shell structure at higher calcination temperature. The results also showed that $SiO_2@TiO_2$ presented higher specific surface area than $TiO_2$ and hollow $TiO_2$ after calcined at 450 °C and 650 °C, this was because the hollow $TiO_2$ calcined at 450 °C and 650 °C could be destroyed by alkaline solution and the etched $TiO_2$ was easy to agglomerate; thus, the core–shell structure was destroyed and the specific surface area decreased. However, the specific surface area of hollow $TiO_2$ calcined at 850 °C and 1050 °C was higher than $SiO_2@TiO_2$ calcined at 850 °C and 1050 °C because the morphology and crystal structure were more stable, the $TiO_2$ shells were not easy to be destroyed at 850 °C and 1050 °C, and the hollow structure was conducive to increasing the specific surface area [28]. The determination results of pore size and pore volume of $TiO_2$, $SiO_2@TiO_2$, and hollow $TiO_2$ were consistent with specific surface area. The results showed that the smaller the pore size, the larger the specific surface area, and the larger the corresponding pore volume. In summary, the calcination temperature greatly affected the specific surface area and mesoporous properties of $TiO_2$, $SiO_2@TiO_2$, and hollow $TiO_2$, and hollow $TiO_2$ showed more obvious mesoporous material properties.

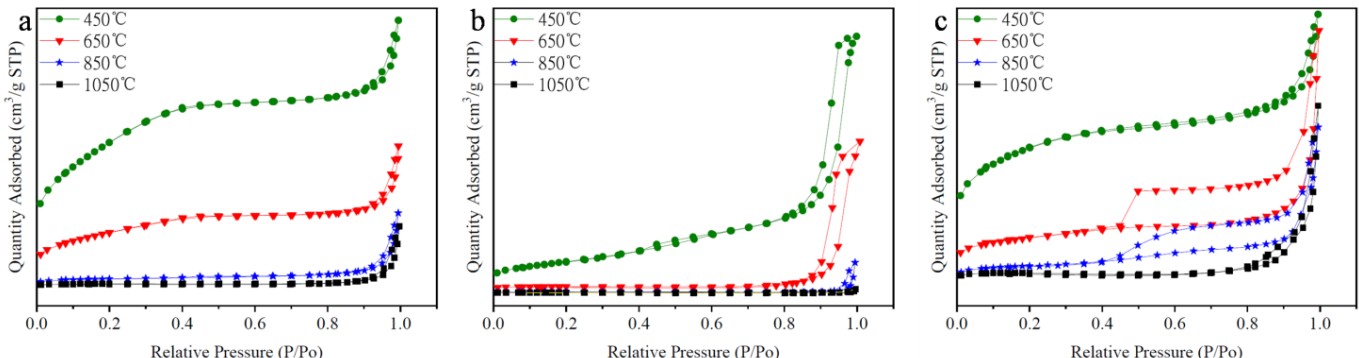

**Figure 7.** $N_2$ absorption-desorption isotherms of (**a**) $TiO_2$, (**b**) $SiO_2@TiO_2$, and (**c**) hollow $TiO_2$ prepared at different calcination temperatures.

**Table 1.** BET specific surface area and fitting parameters of $TiO_2$, $SiO_2@TiO_2$, and hollow $TiO_2$ prepared at different calcination temperatures.

|  | Calcination Temperatures (°C) | BET Surface Area ($m^2g^{-1}$) | Pore Size (nm) | Pore Volume ($cm^3g^{-1}$) |
|---|---|---|---|---|
| $TiO_2$ | 450 | 87.0 | 1.26 | 0.309 |
|  | 650 | 14.4 | 3.85 | 0.164 |
|  | 850 | 2.62 | 6.50 | 0.036 |
|  | 1050 | 0.868 | 1.87 | 0.002 |
| $SiO_2@TiO_2$ | 450 | 280 | 3.19 | 0.256 |
|  | 650 | 105 | 4.12 | 0.120 |
|  | 850 | 14.5 | 25.5 | 0.059 |
|  | 1050 | 5.55 | 62.3 | 0.049 |
| hollow $TiO_2$ | 450 | 213 | 4.84 | 0.157 |
|  | 650 | 73.1 | 1.48 | 0.172 |
|  | 850 | 29.2 | 1.74 | 0.113 |
|  | 1050 | 15.4 | 4.68 | 0.127 |

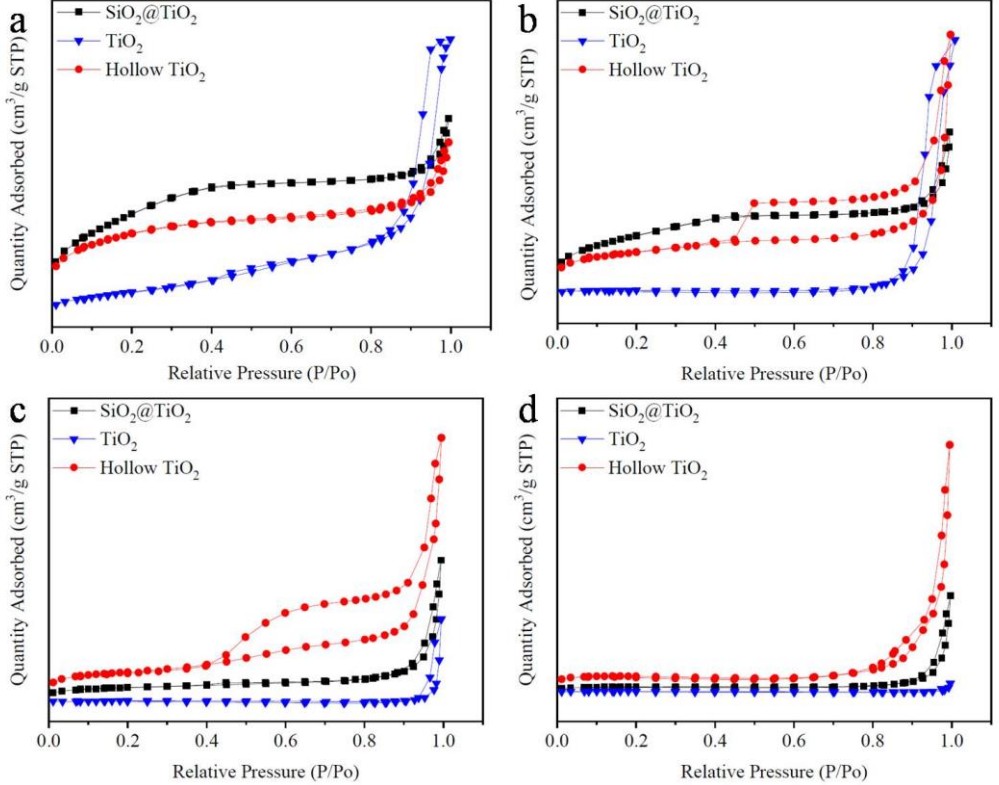

**Figure 8.** $N_2$ absorption-desorption isotherms of prepared nanoparticles calcined at different temperatures (**a**) 450 °C, (**b**) 650 °C, (**c**) 850 °C, (**d**) 1050 °C.

### 3.1.4. Raman Spectra

Raman spectroscopy was also used as an additional characterization method to study the phase transformation of $TiO_2$ at different calcination temperatures for the anatase phase and the rutile phase have different Raman active modes. Figure 9 showed the Raman spectra of hollow $TiO_2$ prepared at different calcination temperatures. As shown in Figure 9, the hollow $TiO_2$ calcined at 450 °C was amorphous due to its unstable crystal structure and did not have Raman characteristic absorption peak. The hollow $TiO_2$ calcined at 650 °C and 850 °C had Raman characteristic absorption peaks at 385 $cm^{-1}$, 505 $cm^{-1}$ and 626 $cm^{-1}$, which belonged to anatase phase Raman characteristic absorption peaks [29]. Additionally, the Raman characteristic absorption peak intensity of anatase calcined at 850 °C was

significantly higher than that of calcined at 650 °C, this was because the crystallinity of hollow $TiO_2$ calcined at 850 °C was higher and the crystal structure was more stable, which was consistent with the XRD results of hollow $TiO_2$ calcined at 850 °C (Figure 1c). The XRD results of hollow $TiO_2$ calcined at 1050 °C showed that it was a mixed crystal phases of rutile and anatase. The Raman spectrum results also indicated that in addition to the Raman characteristic absorption peaks of anatase phase at 385 $cm^{-1}$, 505 $cm^{-1}$ and 626 $cm^{-1}$, rutile phase also had Raman characteristic absorption peaks at 435 $cm^{-1}$ and 601 $cm^{-1}$ [30,31]. The Raman spectra of hollow $TiO_2$ calcined at different temperatures were consistent with the previous XRD characterization results.

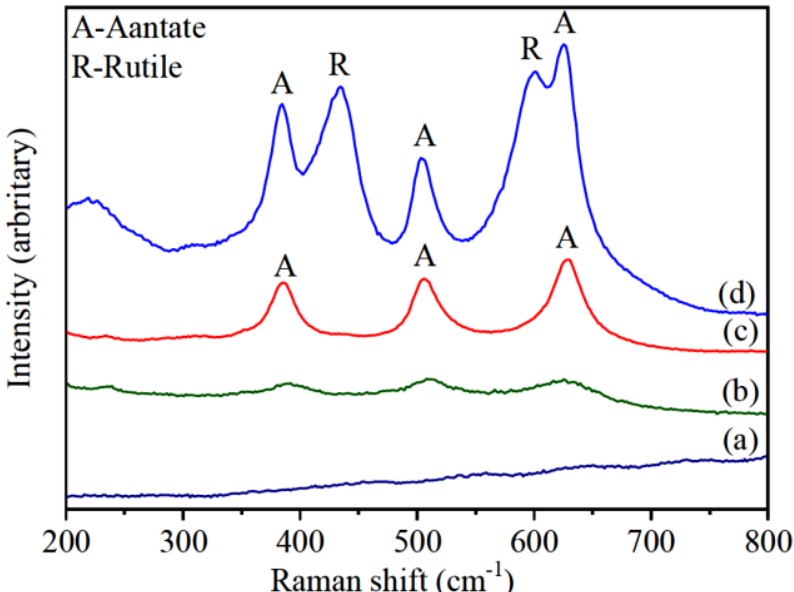

**Figure 9.** Raman spectra of hollow $TiO_2$ at different calcination temperatures (**a**) 450 °C, (**b**) 650 °C, (**c**) 850 °C, (**d**) 1050 °C.

### 3.1.5. Diffuse Reflectance Spectra

Figure 10 showed the UV-vis absorption spectra and the band gap energy of $TiO_2$, $SiO_2@TiO_2$, and hollow $TiO_2$ samples calcinated at 850 °C. The band gap energy was estimated by extrapolating the linear region of the plot of $(\alpha h v)^2$ versus photon energy ($hv$). The optical adsorption spectra of the nanocomposites were shown in Figure 10a, the $TiO_2$ had an absorption band in the region of 380–430 nm because of the crystal structure in rutile phase as shown in XRD results (Figure 1a). The hollow $TiO_2$ showed absorption band around 320–420 nm for the anatase crystal structure as shown in Figure 1c. However, due to the synergistic effect of Ti-Si, the absorption band of $SiO_2@TiO_2$ nanoparticles had a blue shift to higher wavelength region [32]. The calculated band gap energies of $TiO_2$, $SiO_2@TiO_2$ and hollow $TiO_2$ were 2.88, 3.20, and 3.30, respectively (Figure 10b). The lower band gap energy of $TiO_2$ nanoparticles (2.88 eV) was aroused from the rutile phase, the higher band gap width of hollow $TiO_2$ (3.30 eV) could cause a lowering in the energy of valence band and an increase in the conduction band edge, which can promote the separation of electrons and holes and inhibit their recombination, resulting in higher photocatalytic activity [33].

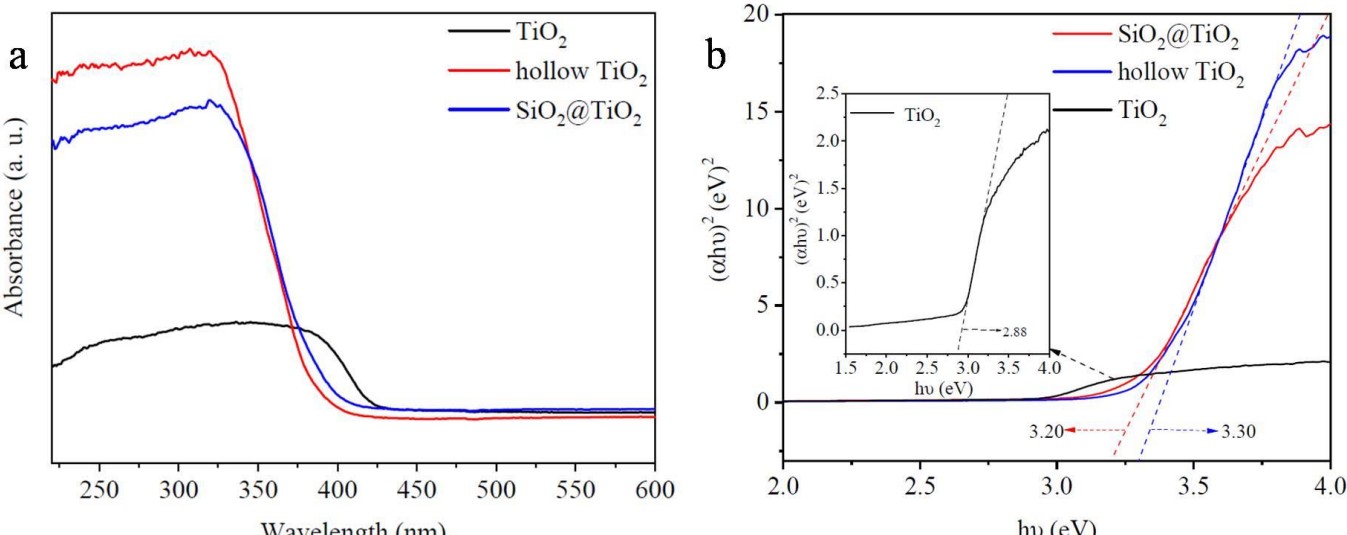

**Figure 10.** (**a**) Diffuse reflectance spectra and (**b**) band gap energy calculation of $TiO_2$, $SiO_2@TiO_2$, and hollow $TiO_2$ prepared at 850 °C.

### 3.1.6. Photoluminescence Spectra

Figure 11 showed the photoluminescence spectra of $TiO_2$, $SiO_2@TiO_2$, and hollow $TiO_2$ samples calcinated at 850 °C. The photoluminescence spectra of semiconductor can provide important information about interfacial electron transfer and the charge carrier recombination process [32]. After excitation at a wavelength of about 400 nm, a broad signal (360–440 nm) was obtained. The results presented that the hollow $TiO_2$ had minimum photoluminescence emission compared with $TiO_2$ and $SiO_2@TiO_2$, which indicated that the thin shell layer of hollow structure reduced the transmission distance of charge carriers and suppressed charge recombination.

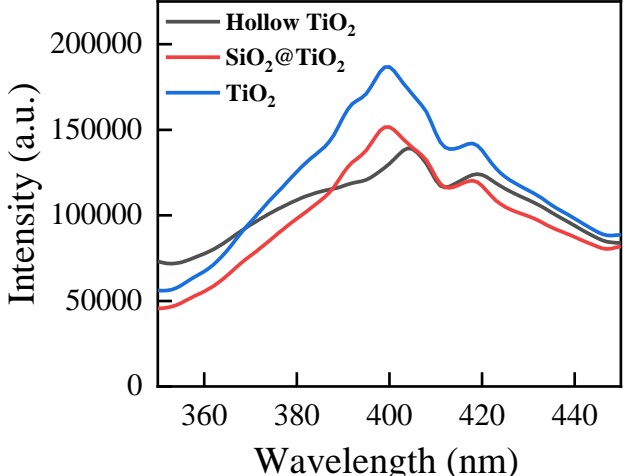

**Figure 11.** Photoluminescence spectra obtained for $TiO_2$, $SiO_2@TiO_2$, and hollow $TiO_2$ prepared at 850 °C.

### 3.2. Photocatalytic Degradation of Phenol

The photocatalytic activities of $TiO_2$, $SiO_2@TiO_2$, and hollow $TiO_2$ prepared at different calcination temperatures for phenol degradation under UV light irradiation were studied in Figure 12. As shown in Figure 12a, $TiO_2$ calcined at different temperatures did not absorb phenol in the dark reaction stage. At the 180 min light reaction stage, the degradation efficiencies of $TiO_2$ calcined at 450 °C and 650 °C were significantly higher than that of $TiO_2$ calcined at 850 °C and 1050 °C, which indicated that the crystal structure of

TiO$_2$ was closely related to photocatalytic activity. The XRD results (Figure 1a) showed that TiO$_2$ nanoparticles calcined at 450 °C and 650 °C were mainly anatase phase, while TiO$_2$ nanoparticles calcined at 850 °C and 1050 °C were mainly rutile phase. It has been reported that anatase TiO$_2$ had better photocatalytic performance than rutile TiO$_2$ [33]. The degradation efficiency of TiO$_2$ calcinated at 650 °C was the highest, 95.2%, which was related to the high crystallinity of anatase TiO$_2$. The photocatalytic activity of SiO$_2$@TiO$_2$ calcined at different temperatures was shown in Figure 12b. The preparedSiO$_2$@TiO$_2$ also had no adsorption on phenol at the dark stage, and the photocatalytic efficiency in the light reaction followed the order of 850 °C > 650 °C > 1050 °C > 450 °C. Compared with phenol degradation by TiO$_2$ calcined at different temperatures, SiO$_2$@TiO$_2$ calcined at different temperatures showed quite different photocatalytic performance due to different phase structure and specific surface area. The crystal structures of SiO$_2$@TiO$_2$ after calcination at 450 °C and 650 °C were anatase phase, but the crystallinity of SiO$_2$@TiO$_2$ was lower than that of TiO$_2$ calcined at 450 °C and 650 °C, as shown in Figure 1b. Therefore, the photocatalytic efficiency of SiO$_2$@TiO$_2$ calcined at 450 °C and 650 °C was lower than that of TiO$_2$ calcined at 450 °C and 650 °C. Additionally, SiO$_2$@TiO$_2$ calcined at 1050 °C also had higher photocatalytic efficiency than TiO$_2$ calcined at 1050 °C for the anatase and rutile mixed crystal structure. However, the SiO$_2$@TiO$_2$ calcined at 850 °C had the best photocatalytic performance of 90.9%, which was due to its high anatase content and high crystallinity. It has been reported that high crystallinity is essential to improve the generation and migration of electron/hole pairs on the surface of TiO$_2$ [34–37].

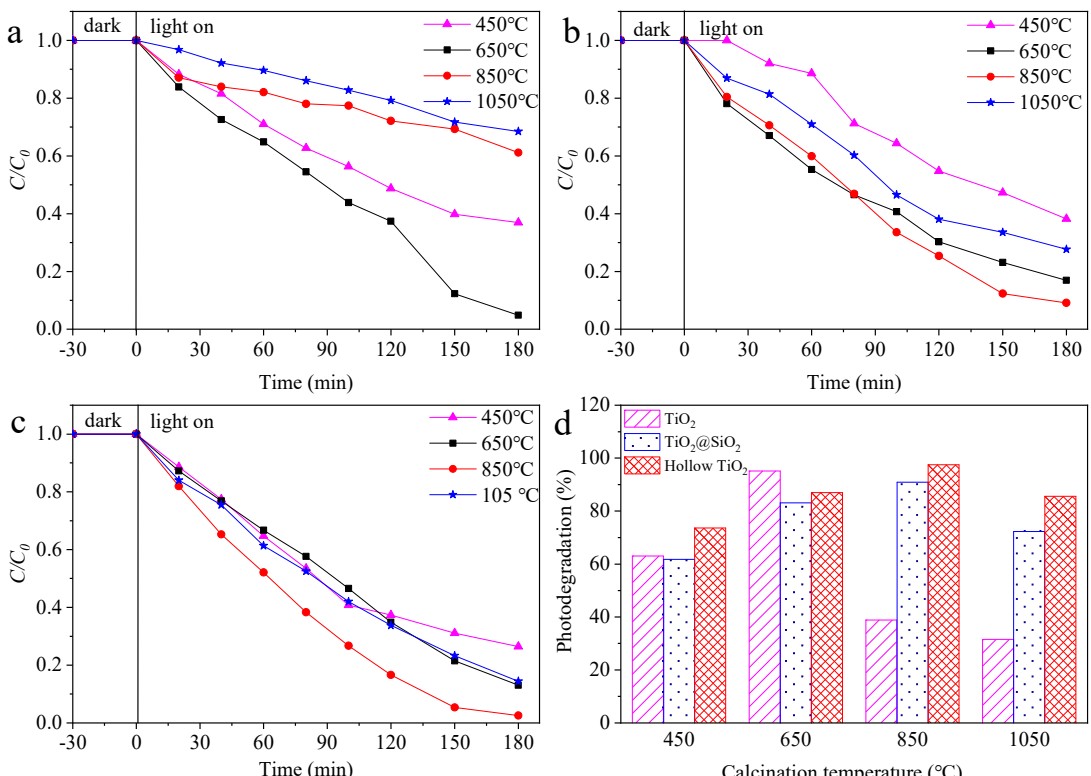

**Figure 12.** Photocatalytic activity of (**a**) TiO$_2$, (**b**) SiO$_2$@TiO$_2$, (**c**) hollow TiO$_2$, and (**d**) photocatalytic degradation efficiency diagram prepared at different calcination temperatures.

Figure 12c presented that hollow TiO$_2$ calcined at different temperatures had better photocatalytic activity than SiO$_2$@TiO$_2$ calcined at different temperatures. The photocatalytic activity of hollow TiO$_2$ calcined at 850 °C was the highest photocatalytic efficiency of 97.5%, which was related to the results of SiO$_2$@TiO$_2$ calcined at 850 °C. As shown in Table 1, hollow TiO$_2$ nanoparticles calcined at 850 °C had larger specific surface area than SiO$_2$@TiO$_2$, which increased light absorption, diffraction and reflection. At the same

time, the thin shell layer reduced the transmission distance of charge carriers and inhibited charge recombination [38]. In addition, the hollow structure increased the dispersion of photocatalysis in the liquid reaction system, supplied more active sites, and improved the photocatalytic efficiency. Figure 12d compared and showed the photodegradation of $TiO_2$, $SiO_2@TiO_2$, and hollow $TiO_2$ prepared at different calcination temperatures for phenol degradation under UV light. In summary, calcination temperature had a great influence on photocatalytic performance. The photocatalytic activities of the three main different nanoparticles abided by the following order: hollow $TiO_2$ calcined at 850 °C > $TiO_2$ calcined at 650 °C > $SiO_2@TiO_2$ calcined at 850 °C > hollow $TiO_2$ calcined at 650 °C > $SiO_2@TiO_2$ calcined at 650 °C > hollow $TiO_2$ calcined at 1050 °C. The photocatalytic activity of $TiO_2$ with different structures mainly depended on the crystal structure and specific surface area. $TiO_2$ with different structures and morphology should control its crystal structure and physical properties to obtain higher photocatalytic efficiency [39,40]. Compared with reported paper about the $TiO_2$-$SiO_2$ and hollow $TiO_2$ composites for photodegradation of phenol as shown in Table 2 [23,36,37,40–42], the hollow $TiO_2$ calcined at 850 °C showed higher photocatalytic efficiency with less concentration of photocatalysis (0.25 g/L).

**Table 2.** A comparison of $TiO_2$-$SiO_2$ and hollow $TiO_2$ composites for photodegradation of phenol.

| Photocatalyst | Synthesis Method | Light Source | Initial Concentration of Phenol (mg/L) | Concentration of Photocatalysis (g/L) | Reaction Time (min) | Efficiency (%) | Ref. |
|---|---|---|---|---|---|---|---|
| $TiO_2$-$SiO_2$ phtocatalyst | Sol-gel | UV light (150 W) | 50 | 1.0 | 120 | 48.0 | [36] |
| Titania-silica composites | Non-aqueous approach | UV light (8 W) | 100 | 3.0 | 180 | 67.0 | [37] |
| $TiO_2$/$SiO_2$ nanoparticles | Hydrothermal method | UV light (30 W) | 10 | 1.0 | 35 | 96.4 | [40] |
| Hollow $TiO_2$-$MoS_2$ | Hydrothermal method | Visble light (300 W Xe lamp) | 10 | 1.67 | 150 | 78.0 | [41] |
| Hollow $TiO_2$ nanocomposite | Sol-gel process | UV light (6 W) | 23.5 | 0.50 | 60 | 21.4 | [42] |
| $TiO_2$-$SiO_2$ hollow nanospheres | Sol-gel method | UV light (175 W) | 80 | 2.5 | 140 | 90.0 | [23] |
| Hollow $TiO_2$ nanocomposite | Sol-gel process | UV light (500 W) | 20 | 0.25 | 180 | 97.5 | This paper |

## 4. Conclusions

The effects of different calcination temperatures on the morphology, crystal structure, specific surface and photocatalytic efficiency of $TiO_2$, $SiO_2@TiO_2$, and hollow $TiO_2$ were deeply and systematically studied. The results verified that $SiO_2$ could inhibit the transfer of $TiO_2$ from anatase phase to rutile phase under high temperature calcination (850 °C). The photocatalytic activity results showed that the photocatalytic efficiency of hollow $TiO_2$ at different temperatures was higher than $SiO_2@TiO_2$. The hollow $TiO_2$ calcined at 850 °C presented the highest photocatalytic efficiency for phenol degradation for the higher crystallinity of anatase structure and specific surface area than $TiO_2$. Therefore, the experimental results showed that the calcination temperature had a great influence on the morphology and photocatalytic activity of $TiO_2$, $SiO_2@TiO_2$, and hollow $TiO_2$. The influence of calcination temperature should be fully considered in the preparation and application of $TiO_2$ photocatalyst.

**Author Contributions:** Methodology, N.F., H.C., R.C. and S.D.; writing—review and editing, X.R. All authors have read and agreed to the published version of the manuscript.

**Funding:** This research was supported by the Yong Scholars Science Foundation of Lanzhou Jiaotong University (2022044).

**Institutional Review Board Statement:** Not applicable.

**Informed Consent Statement:** Not applicable.

**Data Availability Statement:** Data sharing is not applicable to this article.

**Conflicts of Interest:** The authors declare no conflict of interest.

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
