# Peer review of "Effect of Calcination Temperature on the Structure, Crystallinity, and Photocatalytic Activity of Core-Shell SiO2@TiO2 and Mesoporous Hollow TiO2 Composites"

_coatings, doi:10.3390/coatings13050852_

Round 1
Reviewer 1 Report
The article titled "Effect of Calcination Temperature on the Structure, Crystallinity and Photocatalytic Activity of Core-shell SiO2@TiO2 and Mesoporous Hollow TiO2 Composites" has been well-structured and analyzed by the authors. However, the following queries need to be addressed before the article can be accepted:
1. The authors need to explain why studying UV-active photocatalytic systems like TiO2 is still necessary, as stable visible active systems have been extensively studied and have the potential for further exploration.
2. The authors should clarify in what way their study is useful for decreasing the bandgap to the visible region.
3. To ensure the completeness of the photocatalytic activity application, it is recommended that the authors perform a PL study.
4. The authors are advised to cite annealing-based metal oxide semiconductors, such as those in Ceramics International, 46 (2020) 17-22 and Chemosphere 286 (2022) 131577.
5. It is suggested that the authors provide a performance table with other annealed systems for comparison.
Please address these queries before resubmitting the article for consideration.
Author Response
Replies to the comments of the reviewer 1#
Question 1: The authors need to explain why studying UV-active photocatalytic systems like TiO2 is still necessary, as stable visible active systems have been extensively studied and have the potential for further exploration.
Response: Firstly, at present photocatalysis is mainly based on laboratory basic research, and there are only a few types of photocatalysis that can be commercialized and applied in practice, with TiO2 being one of them; Secondly, TiO2 and hollow TiO2 can only use ultraviolet light for photocatalysis, so the light source needs to choose ultraviolet light; Thirdly, although many photocatalysis studies currently attempt to use visible light to reduce the cost of light sources, ultraviolet light is still the strongest natural light source when large-scale photocatalysis is applied to wastewater treatment. Therefore, in the study of photocatalytic wastewater treatment, TiO2 still has certain application prospects by utilizing ultraviolet light.
Question 2: The authors should clarify in what way their study is useful for decreasing the bandgap to the visible region.
Response: The calculated band gap energies of TiO2, SiO2@TiO2 and hollow TiO2 were 2.88, 3.20 and 3.30, respectively (Figure 10b). The lower band gap energie of TiO2 nanoparticles (2.88 eV) was aroused from the rutile phase. Due to the synergistic effect of Ti-Si, the absorption band of SiO2@TiO2 nanoparticles had a blue shift to visible region with band gap width of 3.20 eV. Basically, SiO2@TiO2 and hollow TiO2 had an absorption around UV region.
Question 3: To ensure the completeness of the photocatalytic activity application, it is recommended that the authors perform a PL study.
Response: the PLstudy was supplemented in the article in Figure 11.
Question 4: The authors are advised to cite annealing-based metal oxide semiconductors, such as those in Ceramics International, 46 (2020) 17-22 and Chemosphere 286 (2022) 131577.
Response: The two papers were cited in the paper as shown in
- Babu B.; Shim J; Yoo K. Effects of annealing on bandgap and surface plasmon resonance enhancement in Au/SnO2 quantum dots. Ceram. Int. 2020, 46, 17-22, doi: 10.1016/j.ceramint.2019.08.200.
- Babu B.; Talluri B.; Gurugubelli T.R.; Kim J.; Yoo K. Chemosphere. 2022, 286, 131577, doi: 10.1016/j.chemosphere.2021.131577
Question 5: It is suggested that the authors provide a performance table with other annealed systems for comparison.
Response: A comparison of TiO2-SiO2 and hollow TiO2 composites for photodegradating of phenol was shown in Table 2.
Reviewer 2 Report
Article Effect of Calcination Temperature on the Structure, Crystallinity and Photocatalytic Activity of Core-shell SiO2@TiO2 and Mesoporous Hollow TiO2 Composites
REVIEW:
1. At the end of the Introduction you state … At present, few articles focus…. (add these articles as references!)
2. State the atmosphere during calcination (Part 2.4.)
3. Add the preparation of samples for SEM, TEM characterization - how did you ensure conductivity, transparency on the samples on grids for TEM observation
4. In part 3 – first present the microstructure/morphology of the resulting nanostructures, as you indicate the course (flow) of characterization in Part 2.6
5. Existing Fig. 1 and Fig. 2 combine together in a collage – to make it clearer for readers
6. I suggest the following sequence of existing Figures for the Part 3:
Figure 5 let it be Fig. 1 (SEM),
Figure 3 as Fig. 2 (TEM), Fig. 6 as Fig. 3
Figure 4 as Fig. 4 (HRTEM)
XRD analyses follow - a collage.
This is a logical sequence of presentation of results, the existing presentation of results is chaotic.
7. Conclusions - should be precise and short, do not repeat the content presented in the Abstract
Reviewer 3 Report
Titanium dioxide is an excellent non-toxic photocatalyst and in conjunction with silica offers a composite material with potentially excellent combined properties. The possible applications of this composite are very broad for various surface applications, however, the available information on the properties of this composite is still insufficient and its study is still needed to elucidate all the physical and chemical properties of the material. In the present work, TiO2 photocatalyst with hollow spherical particles was prepared by etching SiO2 core from SiO2@TiO2 composite. Although there is a fairly extensive literature on composites of this type in the current literature dealing with many aspects of their synthesis and applications, I consider the publication of further original results to be useful in view of the overall complexity of the problem. This also applies to the work under review. In the work under review, the focus is mainly on the search for the optimum calcination temperature with respect to the quality of the obtained hollow TiO2 spheres when used as a photocatalyst. The effect of calcination temperature on the microstructure, crystallinity, phase composition and photocatalytic activity of TiO2, core-shell SiO2@TiO2 and mesoporous hollow TiO2 has been described. Furthermore, the correlation between the calcination temperature and the microstructure, crystallinity and photocatalytic performance of the material was sought. Overall, I evaluate the manuscript as interesting and suitable for publication after correcting minor errors and clarifying the correctness of Fig. 7.
· In Figures 1 and 2, the term Aantate is used instead of "anatase".
· Figure 7: subfigures a and c seem to me to be quite identical. Are the images shown incorrect?
Round 2
Reviewer 1 Report
The authors revised the manuscript satisfactorily and answered all questions.
Reviewer 2 Report
OK